# Prognostic and Predictive Value of *PBRM1* in Clear Cell Renal Cell Carcinoma

**DOI:** 10.3390/cancers12010016

**Published:** 2019-12-19

**Authors:** Lucía Carril-Ajuria, María Santos, Juan María Roldán-Romero, Cristina Rodriguez-Antona, Guillermo de Velasco

**Affiliations:** 1Department of Medical Oncology, University Hospital 12 de Octubre, 28041 Madrid, Spain; luciacarril@hotmail.com; 2Hereditary Endocrine Cancer Group, Human Cancer Genetics Programme, Spanish National Cancer Research Centre (CNIO), 28029 Madrid, Spain; msantosr@cnio.es (M.S.); jmroldan@cnio.es (J.M.R.-R.); 3Centro de Investigación Biomédica en Red de Enfermedades Raras (CIBERER), 28029 Madrid, Spain

**Keywords:** polybromo-1, PBRM1, renal cell carcinoma, biomarker, prognosis, predictive role

## Abstract

Renal cell carcinoma (RCC) is the most frequent kidney solid tumor, the clear cell RCC (ccRCC) being the major histological subtype. The probability of recurrence and the clinical behavior of ccRCC will greatly depend on the different clinical and histopathological features, already incorporated to different scoring systems, and on the genomic landscape of the tumor. In this sense, ccRCC has for a long time been known to be associated to the biallelic inactivation of Von Hippel-Lindau (*VHL*) gene which causes aberrant hypoxia inducible factor (HIF) accumulation. Recently, next generation-sequencing technologies have provided the bases for an in-depth molecular characterization of ccRCC, identifying additional recurrently mutated genes, such as *PBRM1* (≈40–50%), *SETD2* (≈12%), or *BAP1* (≈10%). *PBRM1*, the second most common mutated gene in ccRCC after *VHL*, is a component of the SWI/SNF chromatin remodeling complex. Different studies have investigated the biological consequences and the potential role of *PBRM1* alterations in RCC prognosis and as a drug response modulator, although some results are contradictory. In the present article, we review the current evidence on *PBRM1* as potential prognostic and predictive marker in both localized and metastatic RCC.

## 1. Introduction

Kidney cancer is the 12th most frequent solid tumor worldwide with around 400,000 new cases and 175,000 deaths from renal cell carcinoma (RCC) estimated in 2018 [1]. RCC accounts for almost 90% of all kidney tumors, the clear cell (ccRCC) being the most common histologic subtype [2]. 

Approximately 70% of the RCC will present with localized disease, but around one third of them will relapse after surgical excision [3,4]. However, the probability of recurrence will depend greatly on different clinical and histopathological features, which have been incorporated to up to 20 different scoring systems characterized by substantial heterogeneity [5,6,7].

Regarding the molecular characteristics of RCC, the inactivation of the Von Hippel–Lindau (*VHL*) gene is by far the most common oncogenic driver event in ccRCC. *VHL* is an E3 ligase substrate that labels the transcription factors hypoxia inducible factor (HIF) 1alpha and HIF2alpha with ubiquitin for proteosomal degradation. The accumulation of HIF triggers the uncontrolled expression of angiogenesis promoting genes and gives rise to highly vascularized tumors [8]. 

Recent advances in next-generation sequencing (NGS) and their implementation in large cohort of cancer patients (e.g., The Cancer Genome Atlas (TCGA) projects), have led to the identification of additional genes frequently mutated in ccRCC, such as *PBRM1* (≈40–50%), *SETD2* (12%), *BAP1* (10%), and *KDM5C* (5%) [9,10,11]. These are genes coding for proteins participating in the regulation of chromatin remodeling and histone methylation. Although *VHL* is the initiating event in ccRCC, the acquisition of additional mutations, some in subclonal cancer cell populations, is a common characteristic during ccRCC tumor development and metastasis. In this context, multiregion sequencing-based studies have highlighted the potential relevance of *PBRM1* alterations for tumor growth and metastatic capacity, which are determinant for clinical outcome [10]. 

The advances in systemic therapy of advanced RCC (mRCC) in the last 20 years have been remarkable. This progress has led to a significant improvement in the median overall survival (OS), from 13 months in the cytokine therapy era to ≈30 months using combination of targeted therapies and immune checkpoint blockade [12,13,14]. Well established prognostic scoring systems for mRCC, such as the Memorial Sloan Kettering Cancer Center (MSKCC) or the International Metastatic RCC Database Consortium (IMDC), include clinical parameters such as performance status or time from diagnosis to treatment, as well as laboratory parameters [15]. However, other clinical variables which have a prognostic value in mRCC are not included in those models [16]. Likewise, in the current therapeutic context there is a need of updating the scoring systems with molecular parameters, to better personalize treatment strategies to achieve survival improvement and avoid unnecessary toxicities.

## 2. Bromodomain-Containing Protein BAF180 Function

The SWI/SNF complexes are multiprotein machineries able to mobilize nucleosomes and alter chromatin structure through the hydrolysis of ATP molecules. By controlling chromatin accessibility, these epigenetic remodeling complexes act as transcriptional regulators, mediating the binding of transcription factors, coactivators, and corepressors to target promoters and enhancers in the DNA. A wide variety of cellular functions are modulated by SWI/SNF complexes including the balance between self-renewal and cell differentiation, cell cycle, metabolism, and DNA repair processes [17]. SWI/SNF components are recurrently mutated across human malignancies. It is estimated that about 20% of all human tumors harbor alterations in genes encoding SWI/SNF complex subunits [18]. This mutation frequency is comparable to that observed for other canonical cancer genes such as *TP53*, *KRAS*, or *PTEN* [19]. Interestingly, mutations in particular SWI/SNF components are tumor type-dependent, which may indicate that these complexes carry out not only universal but also tissue-specific functions. For instance, *SMARCB1* is found mutated in almost 95% of malignant rhabdoid tumors and *PBRM1* is mutated in ≈40% of ccRCC [20]. Other components of the SWI/SNF complex, such as *ARID1A*, are highly mutated in several cancer types, but rarely altered in ccRCC (e.g., *ARID1A* is mutated in 49% of ovarian clear cell carcinomas, 39% of endometrial cancers, and only in 3% of ccRCC tumors).

The gene *PBRM1* encodes the bromodomain-containing protein BAF180 that serves as the DNA-targeting subunit of the pBAF SWI/SNF complex [21]. BAF180 harbors six bromodomains that bind acetylated residues on histone tails. These domains are capable of recognizing acetylating patterns and target the complete complex to specific chromatin regions. This protein is involved in various DNA repair mechanisms [22] and it is also important for cohesion between centromeres, which is necessary for maintaining genomic stability [23]. 

How mutations in the gene *PBRM1* promote carcinogenesis and tumor progression is still unknown. *PBRM1* is considered a tumor suppressor gene and this role is supported by in vitro experiments in ccRCC derived cell-lines, which show that *PBRM1* gene silencing results in increased proliferation, migration, and colony formation [20]. Consistently, Macher-Goeppinger et al. found that re-expression of *PBRM1* into A704 cells, that harbor a homozygous truncating mutation in this gene, results in diminished colony formation [24]. The role of *PBRM1* as tumor suppressor is also supported by the fact that ≈80% of the somatic mutations found in the gene result in loss of function (LOF) of the protein, not only in ccRCC but also in other tumor types, including breast and pancreatic cancers. Furthermore, BAF180 has been demonstrated to modulate the activity of p53 [24], inducing the transcription of some p53-targets such as p21 and repressing other such as 14–3–3σ (Figure 1). *PBRM1* inactivation can therefore disturb p53-dependent chromatin regulation and enable ccRCC tumors to escape from p53-mediated surveillance [25]. Even if *PBRM1* has mainly been described as a tumor suppressor gene, Murakami et al. recently demonstrated that its role may be context dependent, and that *PBRM1* may also act as an oncogene under specific conditions [26]. They showed that the inactivation of *PBRM1* in 786-O cells, a ccRCC cell line expressing BAF180, resulted in a reduction of cell survival and proliferation. This effect seemed to be dependent on HIF1α/HIF2α expression. In this regard, *PBRM1* acts as a coactivator of both transcription factors, which play opposite roles in established ccRCC tumors. In this particular context, HIF1α acts as a tumor suppressor gene [27], while HIF2α favors tumor development. In in vitro models in which both transcription factors are expressed, *PBRM1* showed tumor suppressive activity. By contrast, when HIF1α was not expressed, *PBRM1* induced expression of HIF2α oncogenic targets, which resulted in increased cell survival and proliferation. In summary, in ccRCC the function of *PBRM1* as a tumor-suppressor or oncogene is context dependent.

## 3. *PBRM1* Mutations in Clear Cell Renal Cell Carcinoma

Hereditary cases of RCC account for 3–5% of all renal cancers. These cases are associated with RCC syndromes and are characterized by young age of onset and multi or bilateral tumors. They are related to mutations in specific susceptibility genes such as *VHL*, *MET*, *FLCN*, *FH*, and *SDHB*, which increase the risk of developing specific RCC subtypes [28,29]. Other minority inherited forms of RCC are associated with *BAP1* tumor predisposition syndrome and constitutional chromosome 3 translocations. Nevertheless, some cases of inherited RCC tumors are not explained by any of the genes mentioned above. Recently, a *PBRM1* mutational screening in a cohort of 35 French unrelated patients with a familial history of RCC revealed one patient with a germline *PBRM1* truncating mutation (p.Asp1333Glyfs). In the family there were three additional cases with ccRCC and the truncating mutation co-segregated with the disease [30]. To our knowledge, this case is the only familial form of RCC explained by a germline mutation in *PBRM1*. This finding could eventually support the inclusion of this gene in genetic screenings for RCC syndromes.

The somatic genomic landscape of ccRCC tumors is characterized by *VHL* inactivation, which is found altered in ≈90% of the cases. Inactivation is typically achieved through the loss of the chromosome arm 3p (encompassing the genes *VHL*, *PBRM1*, *BAP1*, and *SETD2*) in a process of unbalanced translocation followed by the somatic inactivation of the remaining *VHL* allele. However, even if *VHL* biallelic inactivation is a critical founder event, it is not sufficient for tumor development and additional mutations and epigenetic changes are necessary. Extensive sequencing projects by TCGA have identified *PBRM1* as the second most frequently altered gene in ccRCC (49% of explored cases, 163 point mutations, eight deep deletions, and one fusion in a total of 354 cases) [20,31] Even though ccRCC is the tumor type with the highest mutational rate in *PBRM1* in the TCGA Pan-Can Project [32], somatic *PBRM1* mutations were first described in breast cancer [29] and are also observed in other cancer types such as cholangiocarcinoma (22%) and uterine (10%). Notably, the high incidence of mutations in *PBRM1* in renal cancer seems to be limited to clear cell histology and mutations are rare or not observed in other renal cancer subtypes, such as papillary (4%) or chromophobe (0%). In agreement with its role as a tumor suppressor, most mutations observed in ccRCC are truncating (137 out of 163 mutations) while missense mutations are less frequent (23 out of 163). Mutations in *PBRM1* are found distributed along the entire gene, affecting Bromodomains, Bromo-Adjacent Homology (BAH) and High Mobility Group (HMG) domains, with some relevant hotspots of truncating mutations [33] (Figure 2). *PBRM1* gene (at 3p21.1) is also affected by copy number alterations, as a consequence of the characteristic chromosome 3p deletion in ccRCC tumors.

Intratumoral heterogeneity (ITH) is a common feature of primary ccRCC tumors, leading to somatic point mutations and copy number alterations that are only present in a subset of tumor cells within the same tumor [10,34]. Multiregion sequencing showed that renal tumors are characterized by harboring only a small number of clonal mutations, shared by all tumor cells, and a majority of subclonal alterations found at varying frequencies across the tumor. Mutations in *PBRM1* are clonal in ≈75% of cases and subclonal in the remaining 25% of ccRCC primary tumors. Truncal *PBRM1* mutations define particular evolutionary trajectories in the tumor, that impact the prognosis and therapeutic response. Three out of the seven evolutionary patterns described in primary ccRCC tumors presented somatic mutations in *PBRM1* as a truncal event that precede subclonal mutations in *SETD2*, genes from the PI3K pathway or copy number alterations [35] (Figure 3). These three clonal *PBRM1* driven evolutionary patterns are enriched in tumors with high ITH and a large number of different subclones. Tumors following these evolutionary trajectories are characterized by a slow growth rate, are associated with advanced disease and with longer progression free survival (PFS). Consistently, Joseph RW et al. found that the loss of *PBRM1* expression in 1330 ccRCC tumor samples was associated with an increased risk of metastasis development, without an effect on OS [36]. Several studies have described mutual exclusivity between *PBRM1* and *BAP1* mutations in RCC primary tumors [37,38]. Still, truncal *PBRM1* mutations are observed in an additional ccRCC evolutionary subtype characterized by multiple driver clonal mutations in genes that include, in addition to *PBRM1, BAP1*, *SETD2*, or *PTEN*. These tumors have low ITH and high genomic instability and are associated with a rapid progression to multiple sites. *BAP1* clonal mutations are characteristic of an additional ccRCC evolutionary pattern that harbors a high genomic instability and a low ITH. 

Turajlic et al. proposed that tumor clones with mutations in *PBRM1* are preferably selected for metastasis development [35]. However, mutations in additional genes seem to be needed for RCC progression [39]. Metastatic dissemination of primary tumors harboring *VHL* and *PBRM1* mutations as the two only clonal events, is mainly an attenuated process that leads to initial solitary or oligometastases. Metastatic potential in these ccRCC tumors is acquired gradually over time and it is limited to certain subpopulations in the primary tumor that act as a reservoir of metastases. Therefore, mutations in *PBRM1* are a key factor not only for primary tumor development but also for tumor metastatic dissemination, in which additional subclonal mutations, acquired after *PBRM1* loss, seem to play a decisive role. Detection of clonal and subclonal events in *PBRM1*, *SETD2*, PI3K pathway genes, together with somatic copy number alterations is therefore essential to establish a reliable map of the complexity of tumor progression driven by *PBRM1* loss.

## 4. Prognostic Value of *PBRM1* Mutations in Localized and Advanced Disease

Different studies have tried to elucidate the impact of *PBRM1* status in RCC but with controversial results [36,40,41,42,43,44,45,46,47,48,49,50,51]. Two small retrospective studies were conducted to compare the different prognostic implications of both *BAP1* and *PBRM1* mutations in localized ccRCC. The study conducted by Kapur et al. showed no survival differences depending on the *PBRM1* mutational status, but identified two molecular subgroups with different prognostic implications and different gene expression [40]. The *PBRM1*-mutant group showed superior median OS than the *BAP1*-mutant group (HR = 2.7; *PBRM1*-MT 10.6 months vs. *BAP1-*MT 4.6 months). Likewise, Gossage et al. reported worse relapse free survival (RFS) in those tumors carrying *BAP1* mutation when compared to *PBRM1* mutant tumors, although no differences in OS were found (Table 1) [41].

According to recent studies focused exclusively on *PBRM1* status as a prognostic factor, it seems that the *PBRM1* LOF is associated with a different prognostic value in localized and advanced disease.

### 4.1. Localized Disease

Most studies assessing the prognostic value of *PBRM1* in RCC have focused on localized disease. It is noteworthy that only one of these studies has included other histological subtypes aside from ccRCC [42]. The loss of *PBRM1* function has also been correlated with advanced tumor and clinical stage, as well as with more aggressive features, such as lymphovascular invasion and poor differentiation [42,43]. In a systematic review including seven studies (Table 2) *PBRM1* LOF in localized disease was associated with poor prognosis, both in terms of RFS and OS [44].

In 2013 the results obtained by Costa et al. showed that *PBRM1* LOF was associated with poor prognosis and worse 5 years RFS rates (*PBRM1* MT 66.7% vs. *PBRM1* WT 87.3%; *p* = 0.048) and worse 5 years disease-specific survival rates (DSS) (*PBRM1* MT 70.6% vs. *PBRM1* WT 89.7%; *p* = 0.017) [43]. Pawlowski et al. conducted a similar study but including different histological subtypes (227 clear cell, 40 papillary and 12 chromophobe). Their results confirmed that *PBRM1* LOF is more frequent in ccRCC than in other histological subtypes, and suggested that most of truncating mutations in ccRCC negatively affect *PBRM1* expression [42]. The study also showed an association of *PBRM1* LOF with late tumor stage and worse OS (*PBRM1* negative 80 months vs. *PBRM1* positive NR (not reached)). Conversely, a Korean article, which also demonstrated the independent prognostic value of *PBRM1* expression, suggested that the negative prognostic impact of *PBRM1* LOF was limited to lower-stage tumors (stage I/II; *p* = 0.001 for cancer specific survival (CSS)/PFS) and not to higher-stage tumors (stage III/IV; *p* > 0.05), although most of the patients (≈76.6%) were stage I/II [45]. Hakimi et al. investigated the impact of *PBRM1* status in 609 patients with ccRCC from two different cohorts (MSKCC, *n* = 188 and TCGA, *n* = 421) showing no significant impact on CSS. However, the addition of *PBRM1* mutation to *BAP1* mutation, which did correlate with a poor prognosis, increased the risk of cancer death (HR 4.18) [46]. The largest study to date was conducted by the Mayo Clinic and included 1330 patients with ccRCC. *PBRM1* and *BAP1* expression were analyzed by immunohistochemistry (IHC) and they did not find significant differences in CSS according to *PBRM1* expression and adjusting by age [36]. The authors did find, however, a significantly higher risk of metastasis in *PBRM1* deficient tumors. The study was also able to define, with regards to the prognostic significance, four different ccRCC subtypes depending on *PBRM1* and *BAP1* status. Thus, there was a progressive worsening of both RFS and CSS from *PBRM1*+/*BAP1*+ to *PBRM1*-/*BAP1*+ to *PBRM1*+/*BAP1*- to *PBRM1*-/*BAP1*- [36].

Overall, it seems that in localized ccRCC, the loss of *PBRM1* function correlates with aggressive features and advanced stage, as well as with worse prognosis.

### 4.2. Advanced Disease

There is less data on *PBRM1* role in advanced ccRCC compared to localized disease. The first study to assess the prognostic value of *PBRM1* expression in stage IV/recurrent RCC was a Korean study including a small population (*n* = 53) [48]. Patients were divided in two groups using 2.5 IHC score for *PBRM1* expression as the cut-off value. Patients with high expression of *PBRM1* in the tumor had a significantly worse OS than those with low *PBRM1* expression. (HR 2.29, *p* = 0.039), but there was not statistically significant difference when only the ccRCC subgroup was considered (HR 2.22, *p* = 0.068) [48]. Consistently, patients harboring tumors with high expression of *PBRM1* showed shorter OS when treated with both sunitinib and everolimus (low vs. high expression; OS 45.0 vs. 23.0 months, *p* = 0.035) [48]. However, the most robust data come from the study conducted by Voss et al. which included 357 advanced ccRCC patients (COMPARZ trial (training cohort), *n* = 357/RECORD3 trial (validation cohort), *n* = 258) [51]. They confirmed the prognostic role of *BAP1, TP53*, and *PBRM1* in metastatic renal cell carcinoma patients treated with tyrosine kinase inhibitors (TKI). Remarkably, *PBRM1* mutations were associated with a better prognosis, both in terms of PFS (HR 0.67, *p* = 0.004) and OS (HR 0.63, *p* = 0.002) [51], whereas *BAP1* mutations were associated with worse prognosis, and a significant worse OS (HR 1.51). Conversely, Tennenbaum et al. did not find an association between *PBRM1* mutational status (determined by NGS) and OS in 167 advanced ccRCC patients from the MSKCC and TCGA cohorts [50].

Other studies which included cohorts with both localized and advanced disease have shown contradictory results (Table 2) [47,49].

Overall, the studies published to date suggest that *PBRM1* LOF has a different role in localized and advanced disease, constituting a poor prognostic factor in localized disease and a good prognostic factor in advanced disease. The inactivation of *BAP1* and *PBRM1* mainly contribute to different pathways of tumor evolution, being almost mutually exclusive events conferring different prognosis. As suggested, ccRCC could be initiated by a focal mutation in *VHL* followed by 3p loss, predisposing to *BAP1* or *PBRM1* inactivation. The mutation of the remaining allele of *PBRM1* or *BAP1* would lead to tumorigenesis, and depending on which gene is mutated, to different tumor aggressiveness. In advanced RCC, *BAP1* mutation leads to a more aggressive disease and *PBRM1* mutation would be associated with a better prognosis [35].

## 5. Predictive Value of *PBRM1* Mutations 

Antiangiogenics have constituted the cornerstone of mRCC systemic treatment for years. However, during the last years the treatment landscape of mRCC has rapidly changed with the development of new drugs, such as cabozantinib, a multikinase inhibitor, or immune-checkpoint inhibitors, as well as the investigation of different treatment combinations. With such a wide range of treatment options, efforts are now focused on identifying predictive biomarkers of response that will help to establish the best treatment sequence for each tumor and patient. There are studies that suggest the correlation of certain mutations with treatment outcomes [53,54,55]. According to some retrospective studies and preclinical data it seems that the mutational status of *PBRM1* could have a predictive role [48,56,57,58]

### 5.1. Targeted Therapy

Data on *PBRM1* predictive value for targeted therapy is scarce and mainly derive from retrospective studies (Table 3) [48,56], aside from the prospective IMMOTION150 trial [59]. According to the results of the IMMOTION150 study, molecular profiles, related with angiogenesis, immune infiltration, and myeloid inflammation, could constitute predictors of response to immunotherapy and antiangiogenics. Interestingly, the tumors with high angiogenesis were enriched for *PBRM1* mutations. In addition, when comparing the treatment outcomes, there was a clear benefit in sunitinib PFS, compared to atezolizumab monotherapy and to atezolizumab plus bevacizumab, in tumors with *PBRM1* mutations. These results suggest an association between angiogenesis and *PBRM1* mutations, indicating that *PBRM1* mutated patients may benefit more from antiangiogenics than from immunotherapy.

Similarly, Fay et al. performed whole-exome sequencing in pretreatment specimens of 27 mRCC patients who had experienced an “extreme response” (extreme responders or primary refractory patients) to first line VEGF targeted therapy (sunitinib or pazopanib). In agreement with IMMOTION150 study, *PBRM1* mutations were significantly more frequent in patients benefiting from antiangiogenic treatment (54% vs. 7%, respectively) [60].

The retrospective study conducted by Kim et al. investigated the correlation of *PBRM1* expression and treatment outcomes in 53 clear cell mRCC patients [48]. Lines of therapy were not specified. High *PBRM1* expression correlated with worse outcomes in patients treated with either sunitinib or everolimus. When analyzing each drug separately, the effect of *PBRM1* expression was stronger in patients treated with everolimus (median PFS in low and high *PBRM1* expression groups was 3.0 months and 1.9 months, respectively; *p* = 0.10) although it did not reach statistical significance. *PBRM1* expression did not correlate with PFS in patients treated with sunitinib (low vs. high median PFS 7.3 vs. 9.0 months, *p* = 0.83).

Later on, Hsieh et al. investigated the association between *PBRM1* mutational status, analyzed by NGS, and treatment outcomes in 220 clear cell mRCC patients included in the RECORD3 trial [56]. The RECORD3 trial was a phase 2 randomized study comparing sunitinib with everolimus in the first line setting with a crossover design. In this study, patients harboring *PBRM1* mutation had better outcomes than those with *PBRM1* wild type tumors in the everolimus-sunitinib sequence group (MT vs. WT PFS 12.8 vs. 5.5 months; HR 0.53, *p* = 0.004), whereas no differences were seen in the sunitinib-everolimus sequence group (*p* = 0.4).

### 5.2. Immunotherapy

Immune checkpoint inhibitors (ICI) have become a new standard of treatment in RCC. Therefore, there is a need to identify which patients will benefit most from ICI or antiangiogenics in order to better determine the treatment sequence and improve survival outcomes. Although PD-L1 expression is a well established biomarker of response to ICI in some tumors (e.g., non-small cell lung cancer), in RCC it is not useful [59,61,62]. Furthermore, neither the tumor mutational burden or the neoantigen expression have been shown to be suitable for patient selection [63].

The PBAF loss was shown in preclinical *in vivo* studies to increase sensitivity to T-cell mediated cytotoxicity compared to those with intact PBAF [64]. Recently, Miao et al. as part of a prospective trial (CA 209-009), analyzed by whole exome sequencing 35 pretreated tumors [57]. They tried to correlate the tumor genome profile with the clinical benefit from anti-PD1 therapy and discovered there was a correlation between *PBRM1* mutations and clinical benefit. However, when they tried to compare the type of tumor-immune microenvironment in ccRCC tumors according to the mutational status of *PBRM1*, in three different cohorts (TCGA, an independent cohort of untreated ccRCC tumors and patient tumors of their study), tumors harboring *PBRM1* mutations in all three cohorts showed a lower expression of immune inhibitory ligands than those with intact *PBRM1*. 

More recently, a post hoc analysis of the CHECKMATE025 trial which compared nivolumab and sunitinib in ccRCC patients previously treated with antiangiogenics, analyzed the correlation between *PBRM1* status and outcomes (PFS and OS) across-treatment in 382 of the 803 patients included in the trial [58]. The results showed a significant correlation between *PRBM1* status and response to anti-PD1 therapy, with a significant benefit both in PFS and OS in those patients harboring *PBRM1* mutations (HR 0.67, 95% CI, 0.47–0.96; *p* = 0.03, and HR 0.65, 95% CI, 0.44–0.96; *p* = 0.03 respectively) compared to those with *PBRM1* wild type tumors. There was no evidence of an association between *PBRM1* status and clinical outcomes in the everolimus cohort [58]. 

On the other hand, as discussed above, in the clinical trial IMMOTION150 the group of tumors with high angiogenesis showed an enrichment in *PBRM1* mutations and these patients seemed to perform worse when treated with ICI alone (atezolizumab) than when treated with antiangiogenics alone (sunitinib) or in combination with immunotherapy (atezolizumab plus bevacizumab) [59]. 

Altogether, it seems that *PBRM1* mutations are associated with specific molecular tumor expression profiles, and that they may have a predictive role. However, its predictive value is still unclear and needs to be elucidated in future larger randomized trials. 

## 6. Future Perspectives and Conclusions

Cancer is a combination of evolutionary inherited and environmental factors. Most recurrently mutated genes in RCC include *PBRM1*. Evolutionary information from the TRACERX study has shown that RCC tumors with *PBRM1* mutations may have a less aggressive behavior, however, additional events may be synergistic and confer a worse prognosis. This may support that *PBRM1* gene could have dual roles in oncogenesis under different cellular contexts (i.e., localized disease or metastatic disease). This duality in tumor suppressors genes (e.g., *TP53*) have been shown in other tumors [65]. For a gene with both oncogenic and tumor-suppressor potentials, it is possible that one single mutation event would unleash its oncogenic power and abolish its tumor-suppressor function. Theoretically, such mutation events would be easy to target. Indeed, systematic drug sensitivity analyses have proven drug vulnerabilities associated with loss of tumor suppressor genes. This could lead to expanding precision cancer medicine in tumors with a basis on mutations of tumor suppressor genes [66].

In summary, the definition of RCC genetic components that contribute to cancer development is an area of great interest, as it will help to understand cancer evolution and to generate better targeted therapies for the patients. *PBRM1*, the second most commonly mutated gene in ccRCC, seems to play a relevant role shaping tumor aggressiveness and pathway dependencies. This knowledge has important clinical implications and, together with additional genomic markers, will have the potential to personalize the clinical management of RCC patients.

## Figures and Tables

**Figure 1 cancers-12-00016-f001:**
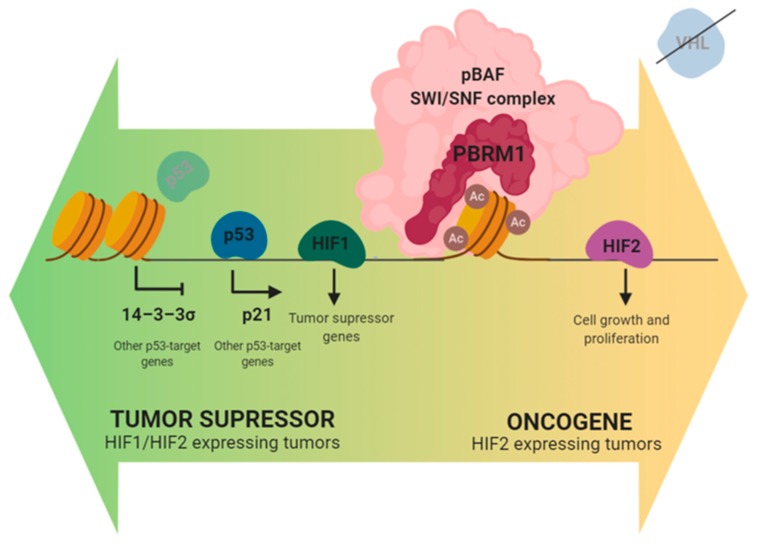
Role of *PBRM1* in established clear cell renal cell carcinoma (ccRCC) tumors.

**Figure 2 cancers-12-00016-f002:**
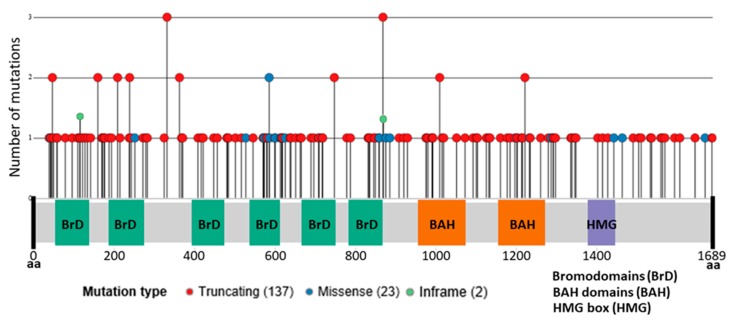
*PBRM1* mutations in clear cell RCC. From The Cancer Genome Atlas (TCGA) PanCancer Atlas Project.

**Figure 3 cancers-12-00016-f003:**
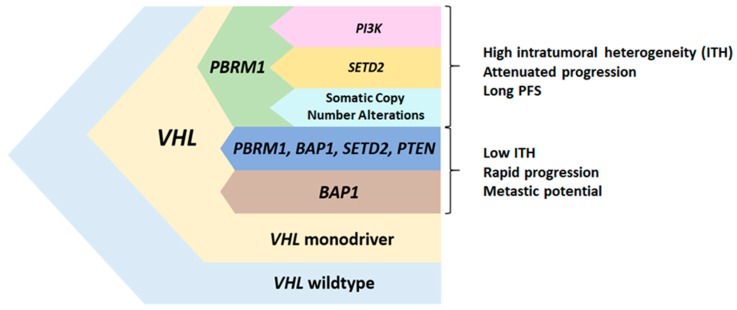
Branched clear cell RCC evolutionary model. Based on Turajlic et al. [35].

**Table 1 cancers-12-00016-t001:** Value of *BAP1* versus *PBRM1* in localized disease.

Author [Ref.]	Year	Country	Gender M(%)/F(%)	Age (y)	*N*	*BAP1* MT(%)	*PBRM1* MT(%)	Tech	Survival Efficacy Parameters	
									RFS (HR/ *p*)	OS (HR/ *p*)
Kapur [40]	2013	USA	80(55)/65(45)	62	145	21(14)	78(54)	NGS	NA	*BAP1* MT: 4.6; *PBRM1* MT: 10.6 (2.7/0.044)
Gossage [41]	2014	UK	83(63)/49(37)	62	132	14(11)	42(33)	NGS	*BAP1* MT: 1.2 ^a^; *PBRM1* MT: 4.9 ^a^ (n.sp./0.059)	n.sp. (n.sp./NS)

M: male patients. F: female patients. Age: median age. y: years. N: number of patients. MT: mutant patients. Tech: Technique. NGS: next-generation sequencing. RFS: recurrence-free-survival in years. HR: hazard ratio. OS: overall survival in years. NS: non statistically significant. NA: not assessed. n.sp.: not specified. ^a^ 75th percentile for survival.

**Table 2 cancers-12-00016-t002:** Current evidence on *PBRM1* prognosis value.

Disease Type	Author [Ref.]	Year	Country	Gender M(%)/F(%)	Age (y)	*N*	*PBRM1* -/Low (%)	Tech	Ab (Dilution)	*PBRM1* Cut-off	Survival Efficacy Parameters	
											RFS/PFS (HR/*p*)	OS/CSS (HR/*p*)
Loc	da Costa [43]	2013	Brazil	66(59)/46(41)	56	112	34(30)	IHC	PB1 Ab ^b^ (1:25)	Presence	*PBRM1* -: 66.7% vs. *PBRM1* +: 87.3% ^d^ (n.sp./0.048)	*PBRM1* -: 70.6 vs. *PBRM1* +: 89.7% ^f^ (n.sp./0.017)
	Hakimi [46]	2013	USA	408(67)/200(33)	61	609	198(33)	NGS	-	-	NA	n.sp. (n.sp./NS)
	Pawlowski [42]	2013	Switzerland	NA	NA	279	175(64) ^a.^	IHC	PB1Ab ^c^ (1:25)	>5%	NA	*PBRM1* -: 80 vs. *PBRM1* +: not reach (n.sp./0.025)
	Nam [45]	2015	Korea	485(74)/172(26)	NA	657	NA	IHC	PB1 Ab ^c^ (1:100)	>50%	*PBRM1* LE: 109.5 vs. *PBRM1* HE: 156.2 (n.sp./<0.001)	*PBRM1* LE: 145.5 vs. *PBRM1* HE: 171.7 (n.sp./<0.001)
	Joseph [36]	2016	USA	823(62)/435(33)	64	1330	674(51)	IHC	PB1 Ab ^c^	Presence	*PBRM1* -: higher mtx. risk (1.46/0.001)	n.sp. (n.sp/NS)
	Kim [52]	2017	Korea	244(70)/107(30)	54	351	208(59)	IHC	PB1 Ab	Score > 2	n.sp. (0.81/0.64)	n.sp. (1.86/0.1)
Loc and Mtx	Jiang [47]	2017	USA	118(74)/42(26)	60	160	49(31)	IHC	PB1 Ab^c^ (1:50)	>5%	n.sp. (0.79/0.21)	n.sp. (0.42/2.45e–05)
	Carlo [49]	2017	USA	77(73)/28(27)	57	105	53(51)	NGS	-	-	*PBRM1* MT: 12.0 vs. *PBRM1* WT: 6.9 ^e^ (n.sp./0.01)	*PBRM1* MT: not reach vs. *PBRM1* WT: 36.3 (n.sp./0.12)
Mtx	Tennenbaum [50]	2017	USA	116(69)/51(31)	60	167	64(38)	NGS	-	-	NA	n.sp. (0.87/0.49)
	Kim [48]	2015	Korea	44(83)/9(17)	62	53	28(52)	IHC	PB1 Ab ^c^ (1:100)	Score >2.5	NA	*PBRM1* LE: 46 vs. *PBRM1* HE: 24 (n.sp./0.022)
	Voss [51]	2018	USA	279(74)/98(26)	61	357	160(45)	NGS	-	-	*PBRM1* MT: 11.1 vs. *PBRM1* WT: 8.2 (0.67/0.004)	*PBRM1* MT: 35.5 vs. *PBRM1* WT: 23.8 (0.63/0.002)

Loc: localized. Mtx: metastatic. M: male patients. F: female patients. Age: median age. y: years. N: number of patients. PBRM1 -: patients not expressing PBRM1. PBRM1 LE: patients with low PBRM1 expression. PBRM1 HE: patients with high PBRM1 expression. Tech: technology. Ab: antibody. IHC: immunohistochemistry. NGS: next-generation sequencing. RFS: recurrence-free survival in months. PFS: progression-free survival in months. OS: overall survival in months. CSS: cancer-specific-survival. MT: mutant patients. WT: wild-type patients. n.sp.: not specified. NA: not assessed. HR: hazard ratio. ^a^: Out of the 175 patients negative for PBRM1 expression 155(68%) were clear cell, 16(40%) papillary and 4(30%) chromophobe histology. ^b^: Anti-BAF180 antibody (Sigma-Aldrich, MO, USA). ^c^: PB1/BAF180 antibody (Montgomery TX, Texas, USA). ^d^: 5-year recurrence free survival rate. ^e^: Time to treatment failure in months. ^f^: 5-year cancer-specific survival rate.

**Table 3 cancers-12-00016-t003:** Current evidence supporting *PBRM1* predictive value for everolimus in metastatic disease.

Author [Ref.]	Year	Country	Gender M(%)/F(%)	Age (y)	*N*	*BAP1* MT (%)	*PBRM1* MT (%)	Tech	Ab(Dilution)	PBRM1 Cut-off	Everolimus PFS	Sunitinib PFS
											*PBRM1* MT or LE vs. *PBRM1* WT or HE (HR/ *p*)	*PBRM1* MT or LE vs. *PBRM1* WT or HE (HR/ *p*)
Hsieh [56]	2016	USA	168 (76)/52(24)	62	220	42(19)	101(46)	NGS	-	-	12.8 vs. 5.5 (0.53/ 0.004)	11.0 vs. 8.3 (0.79/ 0.4)
Kim [48]	2015	Korea	44(83)/9(17)	62	53	NA	25(47)	IHC	PB1 Ab ^a^ (1:100)	2.5	3.0 vs. 1.9 (NA/ 0.1)	7.3 vs. 9 (NA/ 0.8)

M: male patients. F: female patients. Age: median age. y: years. N: number of patients. MT: mutant patients. NA: not assessed. Tech: technique. NGS: next-generation sequencing. IHC: immunohistochemistry. PFS: progression-free-survival in months. HR: hazard ratio. NA: not assessed. ^a^ PB1/BAF180 AB Montgomery, TX.

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
