# Peer review of "Prognostic and Predictive Value of PBRM1 in Clear Cell Renal Cell Carcinoma"

_cancers, 2019, doi:10.3390/cancers12010016_

Round 1
Reviewer 1 Report
The review paper by Dr Carril-Ajuria L et al deals with the current evidence on PBRM1 as potential prognostic and predictive marker in clear cell renal cell carcinoma. I think, most of the hot topics related to PBRM1 were comprehensively and well covered.
However, overall, review article should be easy to follow for readers even without any references. Based on that, I have the following comments.
1. Some of the fonts of Figure 2 and Tables are too small to read. Please re-consider the font and column size.
2. Please put abbreviations carefully for tables and figure 3. MT, ccRCC, HR and NGS in Table 1. AB, mts, exp, ccRCC, pRCC and chRCC in Table 2. MT, WT and 1L in Table 3. ITH and SCNA in Figure 3.
3. Table 1: What does “75th centile” mean? The authors should state in the text or table commentary. There are several blank columns. If they were not assessed in the previous papers, “NA/not assessed” should be stated. The authors did not write p value. Both HR and p values should be included in the table if they want to use “HR/p”.
4. In Figure 2, I could not find blue spots or orange spot which explained inflame mutation or undefined mutation, respectively. These should be highlighted. Full term for “#” should be stated. There was no spot on the line of number 3 or 4. Either explain and justify why these were included, or take them out. This review article was written for PBRM1 in “clear cell RCC”. Therefore, the spots of papillary RCC should be excluded from Figure 2, or make them visible with different colors to be recognized easier.
5. Line 193; Kapur P et al identified two molecular subgroups with different prognostic implications and different “gene” expression, not “genomic” expression according to their paper.
6. Line 253: Not only HRs but also p values should be stated in the text because authors emphasize the article published by H. Voss et al. Similarly, Both HR/p can be stated in Table 2.
7. Line 287: What is the reference (ref)?
8. Line 287, 295, 326 and 327: “PBRM1” should be written by italic. Those are gene name. Please go through the all main text carefully.
9. Similarly, line 113, gene names should be written by Italic.
10. Line 317: Please correct from “stablished” to “established”.
Author Response
1. According to the reviewer suggestion we have now increased both the font and the column size for tables. In addition, we have edited Figure 2 in order to increase its resolution and the size of the labels included. We hope that both tables and Figure 2 are now more understandable and easier to read.
2. Following the reviewer request we have now carefully completed tables’ foot notes with expanded forms for all abbreviations used/included in tables and we have avoided the use of abbreviations in the case of Figure 3.
3. This is a mistake, we meant 75th percentile. This has now been corrected in the manuscript. In addition, we have now filled empty columns with the missing information or with “NA” in the cases for which there is not data available.
4. We agree with the Reviewer and, we have performed several improvements in the Figure 2: First, “inframe” mutations has been located upper from the rest of mutations (truncating and missense). Second, y-axis label is named “number of mutations”. Third, we have adjusted the lines indicating number of mutations (corresponding to y-axis values) to the maximum number of mutations found in one position (three). Lastly, in agreement with the manuscript title, mutations of PBRM1 correspond to those described in cBioportal for Clear Cell Renal Cell Carcinoma, removing 13 mutations from Papillary RCC.
5. We agree with the reviewer that the term “genomic” was not accurate and we have now changed it to “gene”, as suggested.
6. We have now modified the text as requested.
7. This was a mistake. The study was already referenced in previous sentence.
8. We have now gone through the whole text paying attention to what the reviewer points out, we hope we have now corrected all format errors.
9. According to the reviewer advice, we have now corrected it in the main text.
10. According to the reviewer advice, we have now corrected it in the main text.
Reviewer 2 Report
The current review article entitled "Prognostic and predictive value of PBRM1 in clear cell Renal Cell Carcinoma" suggests the potential prognostic and predictive value of PRRM1 gene in ccRCC. I like the overall logical flow to review the current knowledge on RCC and PBRM1 with various aspects of clinical reports. However, I feel that some of information such as the recurrent mutations of PBRM1 in RCC are redundant which limits the clearness of current manuscript.
I suggest two critical things which will help. (1) Could the authors combine section 3 (PBRM1 alterlations in renal cell carcinoma and other cancers) and section 4 (PBRM1 mutation: intratumor heterogeneity and tumor evolution)? (2) The current version of tables is not publication-ready. It is hard to sense the authors' arguments with ease.
Author Response
1. As suggested by the reviewer, we have now combined sections 3 and 4 in a unique section entitled “PBRM1 mutations in renal cell carcinoma and tumor evolution” that gathers the most relevant information included in the two old sections. We have also removed redundant information what we hope makes the text clearer and easier to follow.
2. According to the reviewer’s suggestion we have re-edited the tables and incorporated additional and more complete foot notes in order to make them more understandable and easier to read.